# In Silico Analysis of PORD Mutations on the 3D Structure of P450 Oxidoreductase

**DOI:** 10.3390/molecules27144646

**Published:** 2022-07-21

**Authors:** Muhammad Nurhafizuddin, Aziemah Azizi, Long Chiau Ming, Naeem Shafqat

**Affiliations:** PAPRSB Institute of Health Sciences, Universiti Brunei Darussalam, Jalan Tungku Link, Gadong BE1410, Brunei; 18b3061@ubd.edu.bn (M.N.); 19h8182@ubd.edu.bn (A.A.); long.ming@ubd.edu.bn (L.C.M.)

**Keywords:** cytochrome P450 oxidoreductase (POR), cytochrome P45 oxidoreductase deficiency (PORD), Antley and Bixley’s syndrome (ABS), disorders of sex development (DSD)

## Abstract

Cytochrome P450 oxidoreductase (POR) is a membrane-bound flavoprotein that helps in transferring electrons from its NADPH domain to all cytochrome P450 (CYP450) enzymes. Mutations in the POR gene could severely affect the metabolism of steroid hormones and the development of skeletal muscles, a condition known as Cytochrome P450 oxidoreductase deficiency (PORD). PORD is associated with clinical presentations of disorders of sex development, Antley and Bixler’s syndrome (ABS), as well as an abnormal steroid hormone profile. We have performed an in silico analysis of POR 3D X-ray protein crystal structure to study the effects of reported mutations on the POR enzyme structure. A total of 32 missense mutations were identified, from 170 PORD patients, and mapped on the 3D crystal structure of the POR enzyme. In addition, five of the missense mutations (R457H, A287P, D210G, Y181D and Y607C) were further selected for an in-depth in silico analysis to correlate the observed changes in POR protein structure with the clinical phenotypes observed in PORD patients. Overall, missense mutations found in the binding sites of POR cofactors could lead to a severe form of PORD, emphasizing the importance of POR cofactor binding domains in transferring electrons to the CYP450 enzyme family.

## 1. Introduction

Cytochrome P450 oxidoreductase (POR), previously known as cytochrome c reductase, is an essential membrane-bound flavoprotein that transfers electrons to all the cytochrome P450 (CYP450) enzymes, thus allowing them to fulfill their actions effectively [1,2,3]. Besides providing electrons to all the microsomal P450 enzymes, POR also plays a crucial role in the process of metabolizing drugs, steroid hormone synthesis, and xenobiotics metabolism (Figure 1) [1]. Furthermore, the importance of POR in transferring electrons is not just limited to microsomal CYP450; in fact, other enzymes such as heme oxygenase, cytochrome b5, and squalene monooxygenase, as well as 7-dehydrocholesterol reductase rely on POR to work efficiently (Figure 1) [4]. The gene of POR in homo sapiens contains a total of 16 exons—1 non-coding exon and 15 protein-coding exons—which code for a membrane-bound protein containing 680 amino acids [1]. The POR gene, which is located on chromosome 7q11.23, is anchored towards the endoplasmic reticulum via a hydrophobic N-terminus domain that helps in the interaction with CYP450 enzymes [5].

In terms of its 3D structure, POR comprises three cofactor binding domains, which are the flavin adenine dinucleotide (FAD) binding domain, the flavin adenine mononucleotide (FMN) binding domain, and the NADPH binding domain (Figure 2 and Figure 3) [6,7]. Moreover, the two flavin cofactors (FAD and FMN) are closely connected together via a flexible hinge region that can allow the movement of electrons by undergoing conformational changes, bringing FAD nearer to the domain of FMN (Figure 2 and Figure 3) [5]. The initial step of transferring electrons within POR begins with NADPH being oxidized as it donates its hydride anion (H-) to FAD which will eventually get reduced, the H- ion will then be accepted by FMN, which will pass the electrons to the CYP450 enzyme, specifically the heme center. FAD will receive two electrons from NDAPH but will only transfer one electron to FMN at a time. Similarly, FMN will finally provide the electrons to CYP450 one at a time [4].

### POR Deficiency (PORD)

A number of gene mutations have been reported in the POR enzyme, resulting in the distortion of its 3D protein structure, ultimately causing POR to function abnormally. The mutation-driven changes in the 3D structure of the POR enzyme could lead to a medical condition known as cytochrome P450 oxidoreductase deficiency (PORD) (OMIM: 613571), which is an autosomal recessive genetic disorder and can also be categorized as a rare congenital adrenal hyperplasia (CAH) [3,8]. PORD has an impact on the regulation of steroid hormone production in the body; thus, this condition could affect the normal development of the skeletal system, the reproductive system, as well as metabolic pathways. In an early experiment, Peterson et al. (1985) [9] assessed a 46, XY, six-month-old infant characterized by having a female phenotype and abnormal genitals, with an unusual steroid profile, and it was assumed to be caused by a loss of function of both CYP17A1 and CYP21A2. This report led Miller et al. (1986) [10] to propose that there might be a defect in the POR enzyme, which provides electrons to both CYP17A1 and CYP21A2. In spite of that, the hypothesis was excluded as an embryonic lethality was discovered in POR knockout mice [11,12]. It was not until in 2004, when Flück et al. [3] confirmed that the first four individuals suffering from PORD have mutations in their POR gene. The author also described how three out of four patients developed abnormal genitals while the remaining 1 patient suffered from an abnormal steroid profile, a cystic ovary, and primary amenorrhea [3].

As PORD is a rare autosomal recessive disease about which we have little understanding and knowledge, it can often be misdiagnosed in clinical settings. Missense mutations in the POR gene were seen to be the most prevalent type of mutation causing PORD as compared to other mutations. This study aims to map the missense mutations, reported in PORD patients, on the 3D X-ray crystal structure of the POR enzyme, using a published human POR X-ray crystal structure (PDB: 5FA6). The objectives are to establish a potential correlation between the various POR mutations and their impact on the tertiary structure and function of the POR enzyme, as revealed by observing the clinical phenotypes reported in PORD patients [13,14,15,16].

## 2. Results and Discussion

### 2.1. Compilation of Reported POR Genetic Variants

We have identified a total of approximately 170 cases of PORD reported around the globe since its initial discovery in the year 2004 (Table 1). The extent of the POR enzyme impairment dictates the clinical phenotype, such as disorders of sex development (DSDs), irregular steroid hormone profiles, and skeletal malformations observed in PORD patients. Around 84.1% of PORD patients display ABS features and 70.6% of patients experienced ambiguous external genitalia (Table 1). Furthermore, the tabulated data indicated that a higher number of female individuals with 46, XX karyotype were seen to suffer more from PORD as compared to male, at a ratio of 1.3:1 (Table 1). In addition, a total of 32 reported POR genetic variants, caused by missense mutations in the POR gene, were identified and listed in Table 2, representing 25 disease-causing and 7 polymorphic variants. The identified 32 genetic variants were further mapped on the primary protein sequence (Figure 4) and tertiary structure of the human POR enzyme (Figure 5) using an in silico analysis approach to establish their location within the different cofactor binding domains of the POR enzyme (Table 2).

### 2.2. In Silico Analysis of POR Mutations

To identify the highly conserved amino acid residues in the POR genome across different species, a multiple sequence alignment of human and five POR ortholog proteins (rat, monkey, squirrel, lynx, and dog), sharing above 90% sequence identity, was performed (Figure 6). The sequence analysis identified several highly conserved amino acid residues among the POR orthologs (Figure 6), highlighting the importance of these conserved residues in maintaining the stability of the POR 3D structure and function, as they were retained throughout evolution. The interactions between different amino acid residues within a protein sequence have a great impact on the overall stability of the protein [55], and any change in amino acid residue(s) due to the mutation(s) could drastically influence the structural integrity of the protein [56,57]. The identification of conserved residues within the POR orthologs further prompted us to investigate the location of these conserved amino acid residues on different domains of the POR enzyme in order to understand their potential role in keeping the functional and structural integrity of the POR enzyme.

Therefore, using in silico analysis, we have mapped the 32 POR missense mutant variants on the crystal structure of POR enzyme. The in silico analysis reveals the presence of 9 out of the 32 POR variants on the FMN binding domain (A115V, T142A, G144S, Q153R, G177R, Y181D, N185K, D210G, and P228L) (Figure 5). One of the mutation in the FMN binding domain, i.e., the Y181Dinteracts with the FMN cofactor using a pi–pi bond interaction (Figure 7A) between the benzene ring of the tyrosine group and the isoalloxazine rings of FMN. Therefore, the substitution of tyrosine to aspartic acid at position 181 would result in the destabilization of the FMN cofactor. Several studies have determined that the Y181D mutation leads to a significant loss of activity of cytochrome c and the CYP17A enzyme [1,58,59], leading to bone defects, i.e., midface hypoplasia and phalangeal malformations [13,15]. In contrast, the D210G mutation presents patients with brachydactyly and external genitalia such as micro-penis and hypospadias [14]. The aspartic acid (D210) forms a hydrogen bond with neighboring residues K179 and D218 with an atomic distance of 2.12 Å and 1.76 Å, respectively (Figure 7B), and its substitution to Glycine residue at position 210 would weaken these interactions, resulting in the distortion of the protein structure. As D210G was discovered recently, no POR activity has yet been associated with it.

Furthermore, the analysis of the FAD binding domain reveals the presence of 10 POR mutations (Y326D, L374H, M408L, G413S, R457H, Y459H, R498P, A503V and G504R) (Figure 5). In this group, R457H is the most prevalent mutation found in the Asian population, especially Japanese individuals, presenting with clinical phenotypes varying from mild to severe [3,13,15,16,18,19,20,21,24,25,27,29,41,42,43,47,48,49,53]. The R457 is located at the binding site of FAD where it closely interacts with the FAD cofactor (Figure 7C). The reported in vitro studies showed that the R457H mutation has led to 3% residual activity of the CYP17A1 enzyme [18,60]. In addition, four mutations were located below the FAD-binding domain (A259G, M263V, A287P and R316W) (Figure 5). In this group the A287P mutation commonly found in the Caucasian population accounted for 40% residual activity of POR. The residue A287 plays an important role in stabilizing the beta-sheet structure (Figure 7D), and its substitution with Proline would have deleterious effects on the beta-sheet secondary structure, resulting in severe bone defects and sexual development disorders as seen in PORD patients [3,8,13,15,17,18,22,30,32,35,36,37,40,41].

Similarly, the analysis of the NADPH binding domain also reveals the presence of nine POR variants (G539R, L565P, C569Y, L577R, Y578C, E580Q, Y607C, V608F and H628P) (Figure 5). The mutations in this group, e.g., Y607C, interact with the NADPH cofactor within the NADPH binding domain using a hydrogen bonds and pi–pi interactions (Figure 7E), and its replacement with cysteine at position 607 is reported to have a significant loss in 17,20 lyase and aromatase enzymes activities [59,60], resulting in the reported clinical phenotype of bone deformities and disordered sex developments in both male and female patients [13,27].

## 3. Materials and Methods

### 3.1. Compilation of Reported Cases of POR Mutations

All mutation variants from the reported cases of PORD were listed and tabulated according to their mutated site, mutation residue, DNA change, location on the exon, as well as the domain affected by the mutation. As this study only focuses on missense mutations reported in the POR gene, variants caused by other types of mutations such as frameshift, nonsense, insertion, deletion, duplication, and splice-site mutations are excluded.

### 3.2. Bioinformatics Analysis of POR Protein Sequences

The National Center for Biotechnology Information (NCBI) database was utilized in order to access the Homo Sapiens POR protein sequence (NCBI: NP_000932.3). Once the sequence was retrieved, the computer algorithm basic local alignment search tool (BLAST) (blast.ncbi.nlm.nih.gov; accessed on 25 June 2022) was used to analyze and select POR orthologs from other species that share a close percentage identity with the human POR protein sequences. Based on the BLAST result, species such as Pongo abelli (monkey), Tupaia Chinensis (squirrel), Lynx rufus (lynx), Canis lupus familiaris (dog), and Rattus norvegicus (rat) were selected as the orthologs sharing 99.56%, 94.69%, 94.10%, 93.36%, and 92.48% similarities with the human POR sequence, respectively. Lastly, to find and compare the conserved regions between the human POR sequence and the other selected POR orthologs, the Clustal Omega multiple sequence alignment program (EMBL-EBI, Cambridgeshire, UK) (www.ebi.ac.UK; accessed on 25 June 2022) was used for the alignment of amino acids. Hence, the protein sequences that were used include Homo Sapiens (NCBI: NP_000932.3), Pongo abelli (NCBI: XP_024105393), Lynx Rufus (NCBI: XP_046945282.1), Tupaia Chinensis (NCBI: XP_006155694.1), Rattus novergicus (NCBI: NP_113764.1), and Canis lupus familiaris (NCBI: NP_001171276.1).

### 3.3. In Silico Analysis of 3D X-ray Crystal Structure of POR Protein

A wild-type 3D X-ray crystal structure of POR protein (PDB: 5FA6) was retrieved from the Protein Data Bank (PDB). The Molsoft ICM browser tool (Molsoft LLC, San Diego, CA, USA), was used to analyze the 3D structure of the POR enzyme. In addition, several reported mutations in the PORD patients were analyzed and mapped on the wild-type 3D X-ray crystal structure, and an in-depth in silico structural analysis of the several POR mutations was performed.

## 4. Conclusions

PORD is a genetic steroid disorder with multiple clinical presentations, as observed in patients suffering with PORD. The severity of the PORD clinical phenotypes depends upon the mutated amino acid residue and its location within the three cofactor binding domains, i.e., the FAD, FMN and NADPH binding domains of the POR enzyme. The in silico mapping of missense mutations onto the human POR structure reveals that those amino acid residues that are in direct contact with the cofactor and are substituted for residues with smaller sidechains, e.g., R457 to H and Y181 to D, would result in the distortion of interactions with the cofactor molecule and will compromise the stability and folding of the enzyme. However, a substitution such as A287 to P would interfere with the integrity of a secondary structure element, as the substitution of alanine for proline would introduce a turn/bend in the beta sheet harboring the A287 residue. Patients with the A287P mutation experienced the most severe form of bone defects and sexual development disorders. On the other hand, for residues that display charged interactions within the local environment, such as the two polar residues Y607 and D210 when substituted with cysteine, a non-polar residue, there will be severe steric and electrostatic clashes in the local environment which will severely affect the folding, architecture and stability of the POR enzyme. In contrast, the missense mutations not directly interfering with the cofactor binding domains usually result in less structural destabilization and would present with less severe clinical phenotypes.

## Figures and Tables

**Figure 1 molecules-27-04646-f001:**
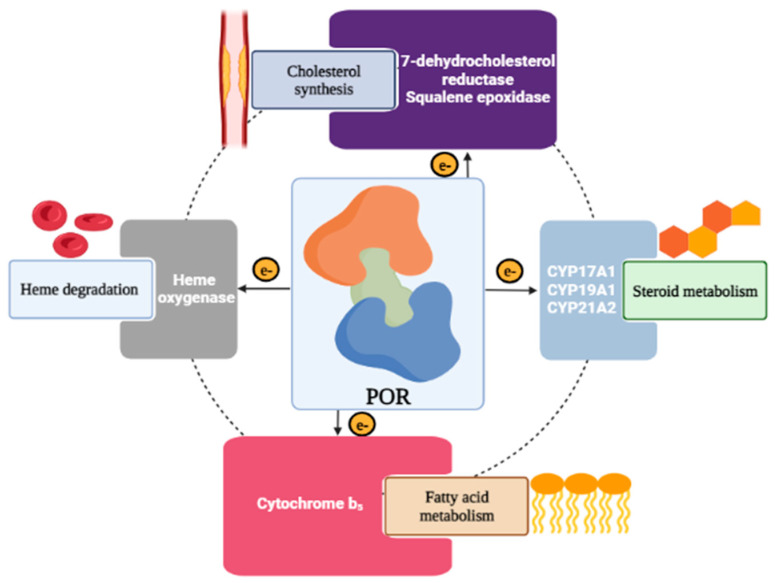
Role of cytochrome P450 oxidoreductase (POR). POR provides electrons to all microsomal P450 enzymes that play a crucial role in the process of metabolizing drugs, steroid hormone synthesis, and xenobiotic metabolism, as well as to other enzymes that also rely on POR to work efficiently, such as heme oxygenase, cytochrome b5, squalene monooxygenase as well as 7-dehydrocholesterol reductase.

**Figure 2 molecules-27-04646-f002:**
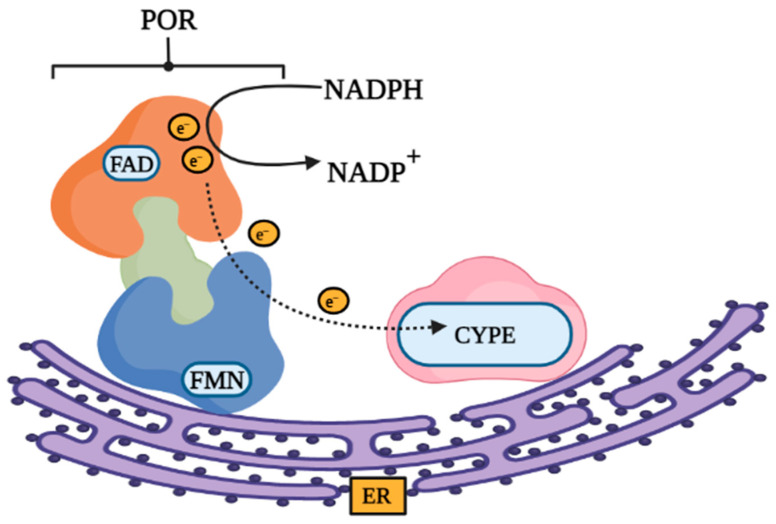
Structure of POR with its redox partner. POR consists of three protein domains which are the flavin adenine dinucleotide (FAD) binding domain, the flavin adenine mononucleotide (FMN) binding domain, and the NADPH binding domain. FAD and FMN are closely connected together via a flexible hinge region that changes shape to help bring FAD towards FMN allowing the electron transfer to take place.

**Figure 3 molecules-27-04646-f003:**
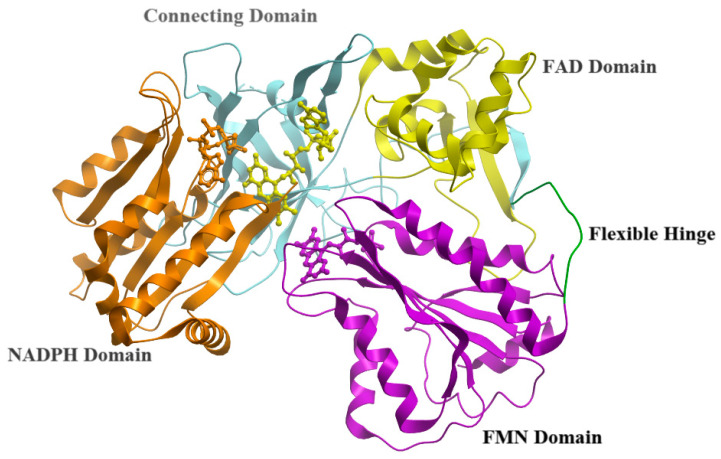
Ribbon model of human wild-type POR protein structure (PDB: 5FA6), displaying the flexible hinge region (green) and the three cofactor binding domains, i.e., FMN (purple ribbon), FAD (yellow ribbon), and NADPH (orange ribbon). The FMN, FAD, as well as NADPH ligands, are indicated as balls and sticks with colors corresponding to their binding domains (purple, yellow, and orange, respectively).

**Figure 4 molecules-27-04646-f004:**
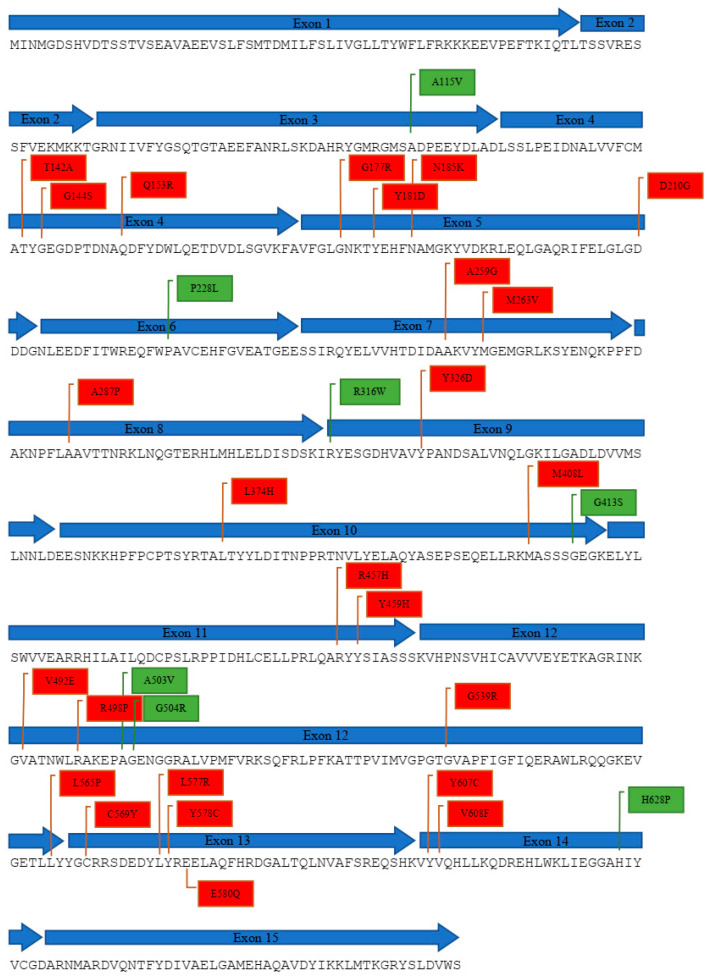
POR protein sequence of Homo Sapiens (NM_000932.3) showing all POR genetic variants due to missense mutations that were reported at their corresponding exon. Red boxes indicate mutation variants found in PORD patients, while green boxes indicate polymorphic variants detected in healthy individuals.

**Figure 5 molecules-27-04646-f005:**
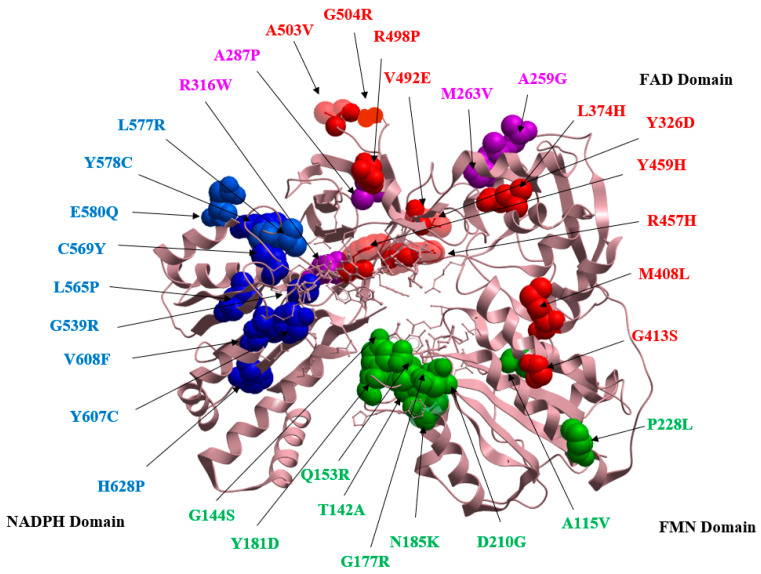
Structure of human POR protein (PDB: 5FA6) represented in a ribbon and balloon model showing the reported missense mutations and polymorphism on their respective binding domains. Green-colored balloons indicate mutations in the FMN domain, red-colored balloons indicate mutations in the FAD domain, and blue-colored balloons indicate mutations in the NADPH domain. Purple-colored balloons indicate mutations found below the FAD-binding domains.

**Figure 6 molecules-27-04646-f006:**
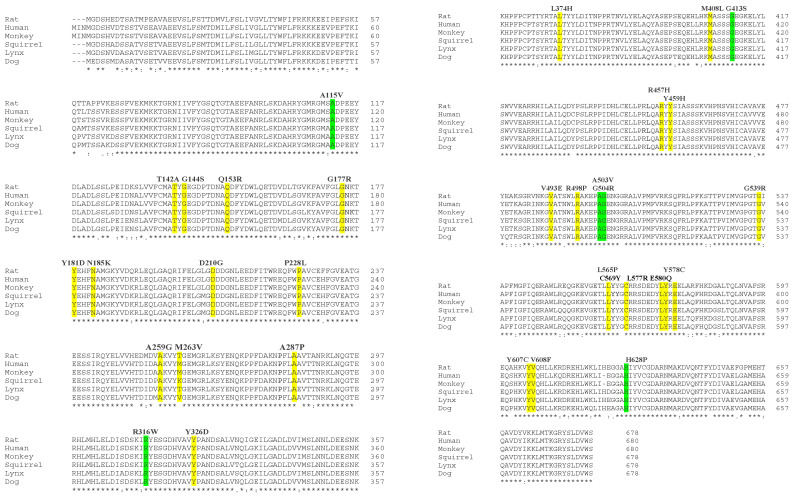
Multiple protein sequence alignment of human and other POR orthologs. The protein sequences used include: rat (NCBI: NP_113764.1), human (NCBI: NP_000932.3), monkey (NCBI: XP_024105393), squirrel (NCBI: XP_006155694.1), lynx (NCBI: XP_046945282.1), and dog (NCBI: NP_001171276.1). The Clustal Omega multiple sequence alignment tool was utilized to make the alignments above. Residues that are highly conserved within the five species are labelled. Polymorphic variants found in healthy individuals are highlighted in green, while the mutation variants found in PORD patients are highlighted in yellow.

**Figure 7 molecules-27-04646-f007:**
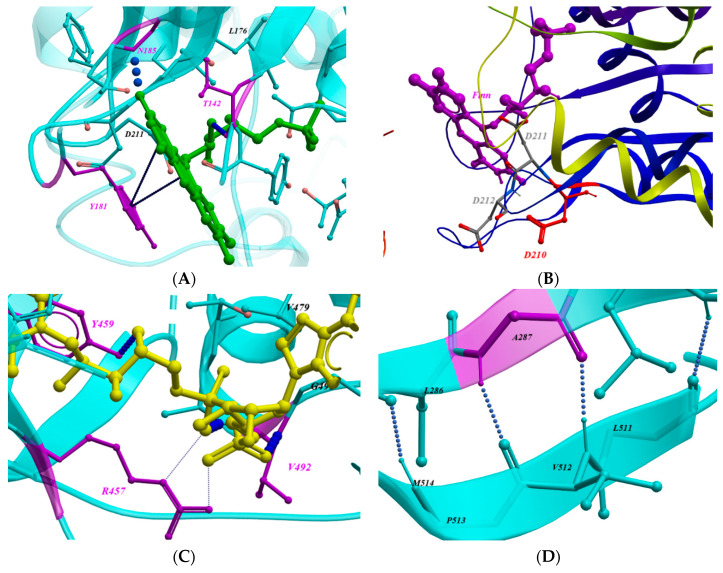
An in silico view of POR cofactor binding domains and location of the conserved amino acid residues. (**A**,**B**) The Y181 and D210 interactions with the FMN cofactor and neighboring residues within the FMN binding domain, respectively. (**C**) The R457 interactions with the FAD cofactor and neighboring residues within the FAD binding domain. (**D**) The A287P interactions with the neighboring residues below the FAD binding domain. (**E**) The Y607 interactions with the NADPH cofactor and neighboring residues within the NADPH binding domain.

**Table 1 molecules-27-04646-t001:** Reported cases of POR deficiency.

Patient (n)	Chromosomal Sex 46,XX/46,XY	POR Mutation	Clinical Phenotype			Reported by
			ABS Features	Abnormal Genitals	Abnormal Steroid Levels	
4	2/2	7/8 ^a^	2	2	3	[3]
3	2/1	6/6	1	2	3	[15]
2	1/1	4/4	2	1	2	[16]
1	0/1	2/2	1	0	1	[17]
19 (32 ^b^)	6 ^c^/10 ^c^	34/38 ^a^	19 (32)	12 ^c^	10 ^c^	[18]
10	6/4	19/20 ^a^	9	9	10	[19]
3	2/1	6/6	0	2	3	[20]
7	2/5	14/14	5	2	7	[21]
1	1/0	2/2	1	1	1	[22]
1	1/0	1/2 ^a,d^	1	1	1	[8]
4	0/4	8/8	3	3	4	[23]
1	1/0	2/2	0	1	1	[24]
12 (35 ^e^)	7/5	24/24	11	11	12	[25]
4	3/1	8/8	3	3	4	[26]
1	0/1	2/2 ^f^	0	1	1	[27]
1	1/0	2/2	1	1	1	[28]
1	1/0	2/2	1	0	1	[29]
7	5/2	14/14	7	5	7	[30]
2	2/0	4/4	2	2	2	[31]
30	18/12	54/60 ^a^	27	22	28 ^c^	[13]
1	1/0	2/2	1	1	1	[32]
1	1/0	2/2	1	1	1	[33]
1	0/1	2/2	0	1	1	[34]
20	12/8	39/40 ^a^	19	12	N/A	[35]
1	1/0	2/2	1	1	1	[36]
1	1/0	2/2	1	1	N/A	[37]
1	0/1	2/2	1	1	1	[38]
1	1/0	2/2	0	1	1	[39]
1	1/0	2/2	1	1	1	[40]
2	1 ^c^/0 ^c^	4/4	1	1	N/A	[41]
1	1/0	2/2	1	1	1	[42]
1	0/1	2/2	1	0	1	[43]
1	1/0	2/2	1	0	1	[44]
1	1/0	2/2	0	1	1	[45]
1	1/0	2/2	1	0	1	[46]
1	0/1	2/2	0	0	1	[47]
1	0/1	2/2	1	1	1	[48]
8	3/5	16/16	7	8	8	[14]
1	1/0	2/2	1	1	N/A	[49]
4	2/2	8/8	2	4	2	[50]
1	1/0	2/2	0	0	1	[51]
2	1/1	4/4	2	1	2	[52]
1	1/0	2/2	1	1	N/A	[53]
2	1/1	4/4	2	0	N/A	[54]
170	95/71		143 (84.1%)	120 (70.6%)		

^a^ Not all alleles were found to be mutated. ^b^ Out of 32 patients, 19 had ABS due to POR mutation. ^c^ Karyotype, description of genitalia, or steroid levels are not known for all patients. ^d^ Additional heterozygous mutations in CYP21B gene. ^e^ Out of 35 patients, 12 were new and other 23 cases were reported earlier. ^f^ Additional heterozygote mutation Q798E in the androgen receptor gene. N/A, Data not available.

**Table 2 molecules-27-04646-t002:** List of missense mutations variants in POR.

No.	Variant	DNA Change	Exon	Domain Affected	References
1	R457H	1370G > A	11	FAD	[3,13,14,15,16,18,19,20,21,24,25,28,30,42,43,44,48,49,50,51,52,53,54]
2	V492E	1475T > A	12	FAD	[3,18]
3	A287P	859G > C	8	Below FAD	[3,8,13,15,17,18,22,30,32,35,36,37,40,41]
4	C569Y	1706G > A	13	NADPH	[3,13,17,18]
5	V608F	1822G > T	14	NADPH	[3,18]
6	A115V	345C > T	5	FMN	[18]
7	T142A	424A > G	4	FMN	[13,18,29,30]
8	Q153R	458A > G	4	FMN	[18]
9	Y181D	541T > G	5	FMN	[13,15]
10	P228L	683C > T	6	FMN	[18]
11	M263V	787A > G	7	Below FAD	[18]
12	R316W	947C > T	9	Below FAD	[18]
13	G413S	1237G > A	10	FAD	[18]
14	Y459H	1375T > C	11	FAD	[18]
15	A503V	1508C > T	12	FAD	[18]
16	G504R	1510G > A	12	FAD	[18]
17	G539R	1615G > A	12	NADPH	[18,23,34]
18	L565P	1694T > C	12	NADPH	[18]
19	Y578C	1733A > G	13	NADPH	[19]
20	E580Q	1738G > C	13	NADPH	[21]
21	L577R	1730T > G	13	NADPH	[26]
22	N185K	555T > A	5	FMN	[26]
23	Y607C	1820A > G	14	NADPH	[13,27]
24	R498P	1493G > C	12	FAD	[13,35]
25	H628P	g.32234A > C	14	NADPH	[13,35]
26	M408L	1223T > A	10	FAD	[33]
27	A259G	758G > C	7	Below FAD	[38]
28	L374H	1121A > G	10	FAD	[39]
29	G144S	430G > A	4	FMN	[45]
30	G177R	529G > C	5	FMN	[46]
31	D210G	629A > G	5	FMN	[14]
32	Y326D	976C > T	10	FAD	[47]

## Data Availability

The data presented in this study are available on request from the corresponding author.

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
