# Peer review of "In Silico Analysis of PORD Mutations on the 3D Structure of P450 Oxidoreductase"

_molecules, 2022, doi:10.3390/molecules27144646_

Round 1
Reviewer 1 Report
Comment 1: In table 2, there are two extra columns “Mutation site” and “Mutated residue”, as the first column “Variant” is self-explanatory.
Comment 2: Authors reported changes in intramolecular interactions with neighboring residues due to mutation. However, it is not clear if all the mutation occurs at the same time or one mutation at a time, since in both cases intramolecular interaction would be different. Moreover, the authors missed mentioning the tool used for generating the mutant model.
Comment 3: Though the present study reports structural changes in the POR structure due to mutation, it would become impactful by studying the effect of inhibitors in mutant structure. Authors may refer to the articles :
• Akhter, M., Tasleem, M., Alam, M. M., Ali, S., “In silico approach for bioremediation of arsenic by structure prediction and docking studies of arsenite oxidase from Pseudomonas stutzeri TS44”, International Biodeterioration & Biodegradation, 2017, Vol.122, page: 82-91. DOI: https://doi.org/10.1016/j.ibiod.2017.04.021.
• Tasleem, M., Ishrat, R., Islam, A., Ahmad, F., Hassan, M. I., “Human Disease Insight: An integrated knowledge-based platform for disease-gene-drug information”. Journal of Infection and Public Health 2016 9(3):P. 331-338 DOI: https://doi.org/10.1016/j.jiph.2015.10.018, PMID: 26631432.
• Tasleem, M., Alrehaily, A., Almeleebia, T.M., Alshahrani, M.Y., Ahmad, I., Asiri, M., Alabdallah, N.M., Saeed, M., “Investigation of Antidepressant Properties of Yohimbine by Employing Structure-Based Computational Assessments”. Curr. Issues Mol. Biol. 2021, 43, 1805-1827. https://doi.org/10.3390/cimb43030127 [I.F. 2.08]
• Saeed M, Saeed A, Alam MJ, Alreshidi M. Computational hunting of natural active compounds as an alternative for Remdesivir to target RNA-dependent polymerase. Cell Mol Biol (Noisy-le-grand). 2021 Jan 31;67(1):45-49. doi: 10.14715/cmb/2021.67.1.7. PMID: 34817369.
• JahoorAlam, M. ., Bardakci, F. ., Anjum, S. ., Mir, S. R. ., Ahmad, I. ., & Saeed, M. . (2022). Computational molecular characterization of p53 and Human TLRs interactions. Cellular and Molecular Biology, 67(5), 1–5. https://doi.org/10.14715/cmb/2021.67.5.1
Author Response
Comments and Suggestions for Authors - Reviewer 1:
Comment 1: In table 2, there are two extra columns “Mutation site” and “Mutated residue”, as the first column “Variant” is self-explanatory.
Respond to the comment:
The two extra columns have been removed from the Table 2, as suggested.
Comment 2: Authors reported changes in intramolecular interactions with neighboring residues due to mutation. However, it is not clear if all the mutation occurs at the same time or one mutation at a time, since in both cases intramolecular interaction would be different. Moreover, the authors missed mentioning the tool used for generating the mutant model.
Respond to the comment:
The reported changes in intramolecular interactions with the neighboring residues are referring to a single gene mutation in P450 gene reported in a PORD patient (as indicated in table 2).
In addition, the program (MOlsoft) used for generating the mutant model has already been mentioned in the methodology part – section 3.3 - In Silico analysis of 3D X-Ray crystal structure of POR protein.
Comment 3: Though the present study reports structural changes in the POR structure due to mutation, it would become impactful by studying the effect of inhibitors in mutant structure. Authors may refer to the articles:
Respond to the comment:
Yes. I completely agree with the reviewer’s advice that by studying the effects of inhibitors in a mutant structure would also be more beneficial. This is something we are planning for our future studies. I would also like to thank reviewer for sharing some important studies to be used as a reference for our next project.
Reviewer 2 Report
Provide reference for the line 92-95.
Figure 2, Keep the abbreviation only. Make ER slightly circular.
In the discussion, please write about any evidence that mutations could also cause change in the stability of proteins.
Author Response
Comments and Suggestions for Authors - Reviewer 2:
Comments 1:
Provide reference for the line 92-95.
Respond to the comment:
The references for the statement on line 92-95 has been added.
Comments 2:
Figure 2, Keep the abbreviation only. Make ER slightly circular.
Respond to the comment:
Figure 2 has been modified. The term “Endoplasmic Reticulum” has been changed to its abbreviation form ER, and ER has been made slightly circular.
Comments 3:
In the discussion, please write about any evidence that mutations could also cause change in the stability of proteins.
Respond to the comment:
A following statement has been added on page 5 (line 124 -127) regarding the impact of mutations on the stability of protein.
"The interactions between different amino acid residues within a protein sequence have a great impact on the overall stability of protein [62] and any change in amino acid residue(s) due to the mutation(s) could drastically influence the structural integrity of protein [63], [64]".
Round 2
Reviewer 1 Report
Accept as it is